# UNAM-HIMFG Bacterial Lysate Activates the Immune Response and Inhibits Colonization of Bladder of Balb/c Mice Infected with the Uropathogenic CFT073 *Escherichia coli* Strain

**DOI:** 10.3390/ijms25189876

**Published:** 2024-09-12

**Authors:** Salvador Eduardo Acevedo-Monroy, Ulises Hernández-Chiñas, Luz María Rocha-Ramírez, Oscar Medina-Contreras, Osvaldo López-Díaz, Ricardo Ernesto Ahumada-Cota, Daniel Martínez-Gómez, Sara Huerta-Yepez, Ana Belén Tirado-Rodríguez, José Molina-López, Raúl Castro-Luna, Leonel Martínez-Cristóbal, Frida Elena Rojas-Castro, María Elena Chávez-Berrocal, Antonio Verdugo-Rodríguez, Carlos Alberto Eslava-Campos

**Affiliations:** 1Laboratorio de Patogenicidad Bacteriana, Unidad de Hemato-Oncología e Investigación, Hospital Infantil de México Federico Gómez, Dr. Márquez No. 162, Col Doctores, Alcaldía Cuauhtémoc, Ciudad de México 06720, Mexico or salcevedom@fmvz.unam.mx (S.E.A.-M.); ricardoeahumada@gmail.com (R.E.A.-C.); joseml@unam.mx (J.M.-L.); frirojc784@gmail.com (F.E.R.-C.); malenachavezb@yahoo.com.mx (M.E.C.-B.); 2Laboratorio de Microbiología Molecular, Departamento de Microbiología e Inmunología, Facultad de Medicina Veterinaria y Zootecnia, Universidad Nacional Autónoma de México, Av. Universidad #3000, Colonia, C.U., Coyoacán, Ciudad de México 04510, Mexico; antoverduro@hotmail.com; 3Unidad Periférica de Investigación Básica y Clínica en Enfermedades Infecciosas, Departamento de Salud Pública, División de Investigación Facultad de Medicina, Universidad Nacional Autónoma de México, Dr. Márquez No. 162, Col Doctores, Alcaldía Cuauhtémoc, Ciudad de México 06720, Mexico; 4Unidad de Investigación en Enfermedades Infecciosas, Hospital Infantil de México Federico Gómez, Secretaría de Salud, Dr. Márquez No. 162, Col Doctores, Alcaldía Cuauhtémoc, Ciudad de México 06720, Mexico; luzmrr7@yahoo.com.mx; 5Unidad de Investigación Epidemiológica en Endocrinología y Nutrición, Hospital Infantil de México Federico Gómez, Dr. Márquez No. 162, Col. Doctores, Alcaldía Cuauhtémoc, Ciudad de México 06720, Mexico; omedina@himfg.edu.mx; 6Laboratorio de Histopatología Veterinaria, Universidad Autónoma Metropolitana Xochimilco, Calzada del Hueso 1100, Col. Villa Quietud, Alcaldía Coyoacán, Ciudad de México 04960, Mexico; olopez@correo.xoc.uam.mx; 7Departamento de Producción Agrícola y Animal, Laboratorio de Microbiología Agropecuaria, Universidad Autónoma Metropolitana Xochimilco, Calzada del Hueso 1100, Col. Villa Quietud, Alcaldía Coyoacán, Ciudad de México 04960, Mexico; dmartinez@correo.xoc.uam.mx; 8Unidad de Investigación en Enfermedades Oncológicas, Hospital Infantil de México Federico Gómez, Secretaría de Salud, Dr. Márquez No. 162, Col Doctores, Alcaldía Cuauhtémoc, Ciudad de México 06720, Mexico; shuertay@gmail.com (S.H.-Y.); bely_16@hotmail.com (A.B.T.-R.); 9Bioterio, Hospital Infantil de México Federico Gómez, Dr. Márquez No. 162, Col Doctores, Alcaldía Cuauhtémoc, Ciudad de México 06720, Mexico; raulcaslu@yahoo.com.mx (R.C.-L.); martinezcristobal@yahoo.com.mx (L.M.-C.)

**Keywords:** urinary tract infections, bacterial lysate, animal model, UTI treatment

## Abstract

Urinary tract infections (UTIs) represent a clinical and epidemiological problem of worldwide impact that affects the economy and the emotional state of the patient. Control of the condition is complicated due to multidrug resistance of pathogens associated with the disease. Considering the difficulty in carrying out effective treatment with antimicrobials, it is necessary to propose alternatives that improve the clinical status of the patients. With this purpose, in a previous study, the safety and immunostimulant capacity of a polyvalent lysate designated UNAM-HIMFG prepared with different bacteria isolated during a prospective study of chronic urinary tract infection (CUTI) was evaluated. In this work, using an animal model, results are presented on the immunostimulant and protective activity of the polyvalent UNAM-HIMFG lysate to define its potential use in the control and treatment of CUTI. Female Balb/c mice were infected through the urethra with *Escherichia coli* CFT073 (UPEC O6:K2:H1) strain; urine samples were collected before the infection and every week for up to 60 days. Once the animals were colonized, sublingual doses of UNAM-HIMFG lysate were administrated. The colonization of the bladder and kidneys was evaluated by culture, and their alterations were assessed using histopathological analysis. On the other hand, the immunostimulant activity of the compound was analyzed by qPCR of spleen mRNA. Uninfected animals receiving UNAM-HIMFG lysate and infected animals administered with the physiological saline solution were used as controls. During this study, the clinical status and evolution of the animals were evaluated. At ninety-six hours after infection, the presence of CFT073 was identified in the urine of infected animals, and then, sublingual administration of UNAM-HIMFG lysate was started every week for 60 days. The urine culture of mice treated with UNAM-HIMFG lysate showed the presence of bacteria for three weeks post-treatment; in contrast, in the untreated animals, positive cultures were observed until the 60th day of this study. The histological analysis of bladder samples from untreated animals showed the presence of chronic inflammation and bacteria in the submucosa, while tissues from mice treated with UNAM-HIMFG lysate did not show alterations. The same analysis of kidney samples of the two groups (treated and untreated) did not present alterations. Immunostimulant activity assays of UNAM-HIMFG lysate showed overexpression of TNF-α and IL-10. Results suggest that the lysate activates the expression of cytokines that inhibit the growth of inoculated bacteria and control the inflammation responsible for tissue damage. In conclusion, UNAM-HIMFG lysate is effective for the treatment and control of CUTIs without the use of antimicrobials.

## 1. Introduction

The increase in antimicrobial resistance is due, among other things, to its inappropriate use in both humans and animals, which has favored the emergence of multidrug-resistant bacteria (MDR) [1]. That is the reason infectious diseases currently represent a major public health problem, which is due to the increasing difficulty in the treatment and control of these [2]. WHO proposed that by the year 2050, a pandemic of catastrophic proportions will be caused by various infectious diseases, for which the institution suggests the development of alternative methods for their treatment and control [3,4,5]. Urinary tract infections (UTIs) are an example of the general impact that infections have acquired in recent years. UTI incidence has increased, affecting populations of different ages, and in many situations, UTIs are associated with other diseases [6,7]. Bacteria are the microorganisms more frequently involved in the pathogenesis of UTIs, and *Escherichia coli* is the main pathogen in both community (70–80%) and hospital acquired infections (60%) [8,9,10]. Strains associated with UTI are known as uropathogenic *E. coli* (UPEC), and they host genes present in plasmids and pathogenicity islands, whose expression contributes to the survival of the bacterium in infected tissues, as well as in the antimicrobial resistance and damage that generates in the host [11,12]. Treatment, in general (preventive and curative), is carried out with antimicrobials, which, in recent years, have decreased their usefulness due to bacterial antibiotic resistance [13,14,15,16]. Resistant bacteria are currently defined as multidrug-resistant (MDR), extremely drug-resistant (XDR) and pandrog-resistant (PDR) [17]. *E. coli* belongs to a bacterial XDR group; thus, WHO included this bacteria in the group of priority microorganisms for the search for alternatives for its control and treatment [4,18]. Vaccines represent one of the main procedures for the control of infectious diseases; however, the great antigenic diversity of bacteria complicates the development of a universal vaccine [19,20]. In UTIs, the participation of different microorganisms complicates the development of a universal vaccine, and, on the other hand, a preventive vaccine is not required since not all of the population is affected [19]. Therefore, the approach to vaccines for UTIs should consider both a prophylactic and a therapeutic treatment [21,22]. Multivalent vaccines manufactured with various bacterial surface components are considered functional biomolecules due to their immunogenic capacity [19,23]. Bacterial lysates described in the last century are compounds with these characteristics [24,25]. In two previous studies, the usefulness of autologous bacterial lysates for the control and treatment of chronic urinary tract infections (CUTI) was evaluated with good results [26,27]. The information obtained on the isolated microorganisms in both studies was used to manufacture a polyvalent bacterial lysate (UNAM-HIMFG). Assays in vitro showed that the compound was both harmless and immunostimulant to culture cells [28]. In this work, the protective effect of UNAM-HIMFG lysate administered sublingually to female Balb/c mice infected with the UPEC CFT073 strain was evaluated. The results strongly suggested that the treated animals with UNAM-HIMFG controlled the infection and the bacterial colonization of the bladder and kidney, stimulating the mice’s immune responses.

## 2. Results

### 2.1. Cultivable Bacterial Microbiota of Mice

The microbiota analysis of Balb/c urine samples showed the presence of *Staphylococcus xylosus* and *Staphylococcus cohnii* subsp. Urealyticum and *Micrococcus luteus*, *Enterococcus gallinarum*, *Listeria seeligeri*, and *Escherichia coli* were found in the feces samples. In C57BL/6 and CD-1, *Micrococcus luteus* and *Lactobacillus acidophilus* were identified in urine samples, and *Micrococcus luteus*, *Enterococcus gallinarum* and *E. coli* in feces. The characterization of *E. coli* isolated from feces did not correspond to the O6:H1 or O25:H4 serotypes, strains used in the infection assays.

### 2.2. Urinary Tract Infection

Balb/c, C57BL/6 mice strains, and CD-1 line were selected for infection assays with UPEC CFT073 (O6:H1) strain. The transurethral infection of C57BL/6 showed the presence of bacteria in urine after the first week of inoculation (5 × 10^5^ CFU/mL); however, from 2 to 5 weeks, a minimal presence of bacteria was observed. Fluctuations in bacterial counts in the following three weeks suggested a recurrent infection process (Figure 1). In CD1 mice, the presence of bacteria (8.3 × 10^3^ CFU/mL) was observed only from the first to four-week post-infection. For Balb/c mice, the presence of bacteria was observed from the first post-challenge week with bacterial counts of 8.0 × 10^4^ CFU/mL, decreasing the two following weeks and increasing to approximately 1.0 × 10^5^ CFU/mL at the end of this study (Figure 1). Considering that CFT073 was a strain isolated from a clinical case of sepsis associated with pyelonephritis, it was decided to evaluate whether the UPEC O25:H4 strain isolated from CUTI could be considered for the infection trial of Balb/c mice. The challenge with this strain only showed the presence of bacteria in the first week (Figure 1). The bladder and kidney of Balb/c and C57BL/6 colonization by CFT703 were evaluated at the end of this study; the presence of bacteria was observed only in the bladder with bacterial counts of 1.5 × 10^3^ CFU/mL.

### 2.3. UNAM-HIMFG Lysate Protects Balb/c Mice against UPEC CFT073 Colonization

Once the conditions of the mice infection and the strain (Balb/c) were established, the protective effect of the polyvalent UNAM-HIMFG lysate was evaluated. The urethral infection assay was performed in different batches of mice with *E. coli* strain CFT073. At 96 h after infection, weekly administration of UNAM-HIMFG lysate or Physiological Saline Solution (PSS) was initiated. The groups of animals challenged with *E. coli* CFT073 treated with UNAM-HIMFG showed a decrease in CFU in urine from one to four weeks, after which no bacteria were recovered (Figure 2). Regarding the animals treated with PSS, the bacterial count remained >10^4^ CFU/mL in all urine samples until the conclusion of this study, with significant differences (*p* ≤ 0.005) compared with animals treated with UNAM-HIMFG lysate (Figure 2). 

### 2.4. Immunostimulatory Effect of UNAM-HIMFG Lysate

TNF-α and IL-10 gene expressions (mRNA) of infected and uninfected (UI) mice treated or not with UNAM-HIMFG lysate were analyzed in the spleen of the animals. The groups CFT073 + UNAM-HIMFG, UI + UNAM-HIMFG, and CFT073 + PSS showed increased TNF-α expression levels when compared to the negative control group (UI + PSS) (*p* < 0.05). TNF-α expression levels did not show significant statistical differences when groups CFT073 + UNAM-HIMFG, UI + UNAM-HIMFG, and CFT073 + PSS were compared against each other. Remarkably, the UI + UNAM-HIMFG animal group expressed higher TNF-α levels (*p* < 0.0001) (Figure 3A). With respect to IL-10, only the UI + UNAM-HIMFG group showed significant statistical differences when compared to the UI + PSS group (*p* < 0.05). However, an increase in IL-10 expression was observed in all groups (Figure 3B).

### 2.5. UNAM-HIMFG Lysate Protects against Tissue Colonization

Histopathological study of the bladder from the control group (UI + PSS) and infected and treated animals (CFT073 + UNAM-HIMFG) did not show histological alterations in the bladder tissue (Figure 4A,B). The same analysis in bladder samples for the CFT073 + PSS group showed leukocytic infiltrate, mild granulomatous inflammation, and the presence of Gram-negative bacteria localized in the submucosa, forming bacterial aggregates (Figure 4C–F). The presence of CFT073 (1.5 × 10^3^ CFU/mL) was only detected in the bladder samples from the CFT073 + PSS group. CFT073 identity was confirmed by culture and PCR detection of lipopolysaccharide and flagellum genes (*wzyO6* and *fliCH1*). No alterations were observed in the histological study of the kidney (data did not show), and bacteria were not isolated in the samples, suggesting no infection in this tissue.

## 3. Discussion

The clinical and epidemiological importance of UTI has increased significantly in recent years, and the MDR is one of the most relevant aspects [1,2,16,29]. A side effect of MDR is that it favors the persistence and chronicity of UTIs, a situation that has begun to attract attention due to the relationship that apparently exists between CUTI, chronic inflammatory diseases, and cancer [30,31]. This has led to the search for alternatives to reduce the use of antimicrobials [32,33], where vaccine development has been considered for the clinical efficacy that they have shown [21,22,34,35]. Multivalent vaccines manufactured with various bacterial surface components are considered functional biomolecules due to their immunogenic capacity [19,23,36], and bacterial lysates described in the last century are compounds with such characteristics [24,25]. In two previous prospective studies of CUTI, the usefulness and efficacy of autologous bacterial lysates for the control and treatment of UTIs were evaluated with satisfactory results [26,27]. Based on these studies, the genera and species of the bacteria that were more frequently identified were selected for the manufacture of the polyvalent lysate UNAM-HIMFG. Assays on cultured cells showed that the compound was harmless and an immunostimulant. [28]. In this work, the properties of UNAM-HIMFG lysate were analyzed using a previously reported mouse infection model [37,38]. The mice infection was carried out with UPEC strains CFT073 and O25:H4; the results showed that UPEC CFT073 and female mice of the Balb/c strain were the most suitable for the proposed study. Additionally, it was observed that UNAM-HIMFG lysate administrated sublingually decreased bladder colonization of CFT073 in Balb/c bladder from the second until the fourth week post-treatment when the presence of bacteria in urine was no longer observed. Unlike what was observed in animals administered with PSS, where the presence of bacteria in urine was identified throughout the sixty days (eight weeks) when the monitoring of the animals was completed (Figure 2). The sublingual UNAM-HIMFG administration route provided good results because it was a route that favored drug absorption, maintained the stability of its components, and eluded enterohepatic circulation [39,40]. It is also known that the administration of antigens through the mucosa activates local and systemic immunity since mucosa-associated lymphoid tissue (MALT) is widely distributed in the nasopharynx (NALT) [nasopharynx-associated lymphoid tissue], intestine (GALT) [gut-associated lymphoid tissue], bronchus (BALT) [bronchus-associated lymphoid tissue], and urogenital tract [41]. On the other hand, in the mucosal immune system, M cells found in the epithelium belonging to GALT and NALT have an important role in immunization since they recognize bacterial structures as lipopolysaccharide (LPS), FimH, and peptidoglycan [42,43]. The characterization of UNAM-HIMFG lysate carried out previously showed that LPS, peptidoglycan, and various protein components were found in high concentrations in the lysate [28]. Therefore, the route of administration used and the composition of UNAM-HIMFG favor the local and systemic immune response [41]. It is known that MALT has different locations, which are connected by the common mucosal immune system (CMIS) through the migration of lymphocytes induced by a specific antigen in one site of the mucosa as effector cells to different organs and protect the tissue against infection [44]. The secretion of TNF-α in human macrophages and murine cell cultures induced by UNAM-HIMFG previously reported [28] was confirmed in the present study in the animal model (Figure 3A). TNF-α has been shown to act as an innate helper factor produced by Ly6C macrophages+, whose function is to favor chemokines secretion (CXC motif) ligand 2 (CXCL2), responsible for the increase in the inflammatory response and neutrophile migration to the infection site [45,46,47]. The LPS, both for its concentration and diversity, is an important component in the UNAM-HIMFG lysate [28]; functionally, this molecule acts as an adjuvant during immune recognition, which, in turn, stimulates IκB kinase, responsible for eliminating factors that inhibit the expression of NF-κB, a mediator that favors the transcription of genes associated with proinflammatory molecules such as TNF-α [48,49,50,51]. The above suggests that the high LPS content of UNAM-HIMFG lysate favors the TNF-α gene expression, which is higher in treated animals (Figure 3A). The IL-10 expression (Figure 3B) was observed in both treated and untreated animals that have been administered with UNAM-HIMFG lysate; this interleukin, due to its anti-inflammatory effect, is produced to regulate the microenvironment of an infectious process such as UTIs [52,53,54]. The anti-inflammatory effect induced by UNAM-HIMFG associated with IL-10 expression was confirmed in the histological study of not treated UNAM-HIMFG animals (Figure 4C,D), in which an intense inflammatory reaction was observed, suggestive of a chronic process [30,31]. In contrast, the inflammatory reaction did not occur in the control animals and in those that received the lysate (Figure 4A,B). This observation suggests that, although the animals were colonized by CFT073, the inflammatory event associated with the infection was probably controlled by IL-10 gene activation induced by the UNAM-HIMFG lysate. 

As mentioned above, the control and treatment of CUTI with antimicrobials is becoming more difficult every day due to the emergence of resistant mutants that influence changes in sensitivity profiles, among other factors. An example of the above is the recent reports by Bologna et al. [15] and Cai et al. [16], who observed an increase in fosfomycin resistance in strains producing extended spectrum beta lactamases and the existence of strains of the Enterobacteriaceae family that are carbapenem-resistant, respectively. On the other hand, the recurrence and chronicity of UTIs associated with the difficulty that already exists for their control and treatment lead to considering a proposal that has existed for more than 40 years regarding the association between CUTI and bladder cancer [55]. In this sense, recent studies propose that urinary infections participate in bladder carcinogenesis. In this regard, it is suggested that prolonged inflammation contributes to tumor development and that proinflammatory cytokines play an important role with TNF-α as the possible molecular link between chronic inflammation and cancer [56]. T Hannan TJ et al. [30] observed that at the beginning of the infection, the innate immune response activates a cascade of proinflammatory cytokines that triggers a severe acute inflammatory response, which causes significant tissue damage. In the present work, it was also observed that infection of mice with a UPEC strain activated the expression of TNF-α; however, the administration of a polyvalent bacterial lysate (UNAM-HIMFG) controlled both colonization and inflammation (activating the expression of IL-10) of the tissues. Although new infection assays with longer follow-ups of the animals are required, it can be suggested that lysate prevents the occurrence of a chronic inflammation event and could be evaluated for the control of bladder cancer. Another aspect with equal relevance to those previously mentioned is the specific function of the urinary tract microbiota and the action of antimicrobials causing dysbiosis and how both can participate in the control or development of bladder cancer [31,56].

## 4. Materials and Methods

### 4.1. Ethics Statement for Animal Protocols

All experiments were conducted according to the specific techniques for the production, care, and use of laboratory animals described in NOM-062-ZOO-1999 [57] and approved by the Institutional Subcommittee for the Care and Use of Experimental Animals (SICUAE) [SICUAE.DC2020/3-1], the Faculty of Veterinary Medicine and Animal Husbandry UNAM, the Internal Committee for the Care and Use of Laboratory Animals (CICUAL) [CICUAL 002-CIC.2021] of the Faculty of Medicine UNAM, the Ethics Committees: Research Division, Faculty of Medicine UNAM [FM/Dl/013/2021], and Research, Ethics and Biosafety, Hospital Infantil de México Federico Gómez [HIM-2023-008].

### 4.2. Bacterial Strains and Culture Conditions

UPEC O25:K2:H4, a clinical isolate from a patient with chronic UTI; UPEC E. coli CFT073 (ATCC 700928) (O6:K2:H1). Culture media: blood agar [BA] Bioxon, Cuatitlán Iz-calli, Edo. Mex., México), MacConkey agar [McC] (Bioxon, Cuatitlán Izcalli, Edo. Mex., México); Cystine Lactose Electrolyte Deficient [CLED] agar (Oxoid, Basingstoke, Ham., England); Thioglycollate broth with dextrose and indicator (Bioxon, Cuatitlán Izcalli, Edo. Mex., México), Luria broth [LB] (Dibico, Cuatitlán Izcalli, Edo. Mex., México). Anaerobic agar [AN]; Brucella agar (BD Difco, Cuatitlán Izcalli, Edo. Mex., México) with 5% defib-rinated ram blood; 1 g/L sodium pyruvate (Sigma-Aldrich, St. Louis, MO, USA); 10 mg/L vitamin K, 400 mg/L sodium thioglycolate (Sigma-Aldrich, St. Louis, MO, USA.); 400 mg/L L-cystine (Sigma-Aldrich, St. Louis, MO, USA). In addition, for the selective isolation of Gram-negative anaerobic bacteria, 7.5 mg/L vancomycin (Sigma-Aldrich, St. Louis, MO, USA) was added [ANV]. For the selective isolation of Gram-positive anaerobic bacteria, 10 mg/L nalidixic acid (Sigma-Aldrich, St. Louis, MO, USA) was added [ANN].

### 4.3. Laboratory Animals

Balb/c and C57BL/6 mice strains and CD-1 line were provided by the bioterium Unit of the Hospital Infantil de México Federico Gómez. The animals were housed in the same unit under sterile conditions in a rack with airflow of 20 changes per hour, relative humidity of 45 to 65% at a temperature between 18 and 22 °C, and wooden substrate bedding; they were hydrated and fed with sterile water and food.

### 4.4. Cultivable Microbiota of the Mice

Prior to animal infection, urine and fecal cultures were performed [58]. Samples were collected in sterile tubes (Axygen, Union City, CA, USA); feces were suspended in sterile Physiological Saline Solution (PSS). The urine was collected directly in sterile tubes (Axy-gen, Union City, CA, USA), and both samples were cultured in ANV and ANN and placed in GENbag anaerobic bags (Biomérieux, Marsella, France) to anaerobic bacteria; BA and McC were used for aerobic cultures. Isolated anaerobic colonies were transferred to thio-glycollate broth for identification in VITEK MS (Biomérieux, Marsella, France); aerobic colonies were identified by metabolic tests [59].

### 4.5. Inoculum Preparation and Conditions

Inoculum preparation of CFT073 and O25:H4 UPEC strains was performed using two culture conditions: (1) static liquid medium (LB) and (2) BA, as previously reported (37). Colony Forming Units (CFU) present in the inoculum were determined by plate counting (CFU/mL) using the drop-seeded technique on BA plates. To stimulate the expression of F9 adhesin, the inoculum was incubated at 25 °C for 20 min, as previously suggested [60].

### 4.6. Urinary Tract Infection in Murine Model 

Six female (6 to 8 weeks age) Balb/c, C57BL/6, and CD-1 mice strains from each group were challenged with UPEC strain CFT073 to evaluate which mice strain responded better to infection. The animals were anesthetized intraperitoneally with 2% Xylazine HCl (Xylazine, Aranda, Edo. Mex., México) and 10% Ketamine (Ketamin Pet, Aranda, Mex., México) at 5 mg/kg and 60 mg/kg doses, respectively. Once the surgical plane was confirmed, the mice were infected with 25 µL of the standardized culture (1 × 10^8^ CFU/mL) using sterile catheters (for intravenous administration) 24G caliber (Punzocat Vizcarra, Edo. Mex., México). The catheter introduction was made through the urethra until reaching the bladder (approximately 3 mm posterior to the urinary meatus); control animals were administered PSS with the same maneuver [38,61]. The animals were observed for 60 days, with urine samples being taken weekly until the conclusion of this experiment [58]. Urine samples were plated on McC or CLED and BA; the bacteria identified as *E. coli* were subjected to DNA extraction and PCR assays, using primers to amplify fragments of *wzy* and *fliC* genes (Table 1), O6 and O25, and H 1 and H4 somatic and flagellar antigens, respectively [62,63].

### 4.7. Protective Effect of UNAM-HIMFG Lysate 

Four groups with Balb/c mice strains were integrated (n = 10) using UPEC CFT073 for infection and UNAM-HIMFG for protection: (1) CFT073 + UNAM-HIMFG; (2) CFT073 + PSS, (3) UI + UNAM-HIMFG; (4) UI + PSS. Urine samples were collected before and weekly after infection until completing 60 days [38,58]. For counting and isolation of bacteria, 5 µL of urine was diluted to obtain a final volume of 50 µL to inoculate culture plates of BA and CLED. Once the infection was confirmed, the identification of bacteria was performed, as described above (Section 4.6). UNAM-HIMFG lysate was prepared, as previously described [28], and administered sublingually with 1.0 inch long 22 G-gauge blunt-tipped straight metal esophageal cannula (sterile) connected to a sterile 1 mL plastic syringe [39,41]. The cannula was passed through the diastema, and the tip was placed under the tongue; the lysate administration was at 20 to 30 s intervals until a volume of 0.2 mL was collected; this treatment was performed twice each week for 60 days [40,64,65]. 

### 4.8. Cytokines Detection 

On the 60th day, mice were anesthetized with xylazine with ketamine (as was described in Section 4.6) and euthanized by cervical dislocation according to NOM-062-ZOO-1999 [57]. Spleens were collected from the animals in sterile 0.6 mL tubes (Axygen, Union City, CA, USA) and stored in an ultra-freezer (Revco, Thermo Fisher Scientific, Ashevie, NC, USA) at −70 °C until processing. Each spleen was macerated by placing it on a cell strainer (CELLTREAT, Scientific Products, Pepperell, MA, USA); then, a volume of 600 µL of PSS was added, and with a 5 mL syringe plunger, the tissue was disaggregated on Petri dish. The macerate tissue was collected in sterile conditions in 1 mL tubes (Axygen, Union City, CA, USA), and mRNA extraction was performed using the PureLink RNA Mini Kit (Invitrogen, Carlsbad, CA, USA) following the manufacturer’s instructions. The mRNA was quantified by spectrophotometry (NanoDrop, Thermo Fisher Scientific, Asheville, NC, USA), and then 10 µg of mRNA was treated with 2 U DNase (Thermo Scientific, Vilnius, Vilnius county, Lituania). The mixture was incubated for 10 min at 25 °C in a thermocycler (MiniAmp Thermal Cycler, Applied Biosystems, Thermo Scientific Fisher, Waltham, MA, USA); then, 2 mM EDTA to inactivate DNase I (65 °C for 15 min) was added. The mRNA was processed by retro-transcription with QuantiTect Reverse Transcription Kit (Qiagen, Hilden, North Rhine-Westphalia, Germany) following the manufacturer’s instructions. With the cDNA, the qPCR was performed with the SoAdvanced Universal SYBR Green Supermix kit (Bio-Rad, Hercules, CA, USA) and the respective primers (Table 2). The expression levels of TNF-α and IL-10 genes were measured using a fragment of the glyceraldehyde 3-phosphate dehydrogenase (GAPDH) gene sequence as a constitutive gene [47,66,67]. The reactions were carried out on the AriaMx real-time PCR System thermal cycler (Agilent Technologies, Santa Clara, CA, USA), and the results were analyzed by means of the 2^−ΔΔCT^ method [68].

### 4.9. Effect of UNAM-HIMFG Lysate on Tissue Colonization by Bacteria 

The protective effect of UNAM-HIMFG lysate on the establishment of bacteria in bladder and kidney samples from the mice studied was analyzed. Kidneys and bladder were collected aseptically; one kidney from each animal and a portion of their bladder were placed in sterile microtubes (Axygen, Union City, CA, USA) and fixed with 10% formaldehyde (J.T. Baker, Xalostoc, Edo. Méx. México) buffered to pH 7.0–7.4. To evaluate tissue alterations, these were embedded in kerosene, sectioned, and stained with Hematoxylin and Eosin [69] and Sandiford’s stain [70]. Kidneys and the bladder portions that were not fixed were macerated, as was previously mentioned (Section 4.8); dilutions were performed in PSS, and samples of these were inoculated in BA and CLED to determine CFU/mL.

### 4.10. Statistical Analysis

Statistical analysis was performed using GraphPad Prism version 10 software (GraphPad Software, San Diego, CA, USA). UPEC CFT073 prevalence in Balb/c mice was compared by Fisher’s exact test. TNF-α and IL-10 expression between groups were compared by performing a Student’s *t*-test. For all analyses, a value of *p* < 0.05 was considered statistically significant.

## 5. Conclusions

Results showed that the CFT073 UPEC strain colonized the bladder of Balb/c mice for a reasonable time for a CUTI study. With respect to the UNAM-HIMFG lysate, it was observed that it had a protective effect on bacterial proliferation, eliminating them, avoiding tissue lesions, and controlling the chronicity of the clinical manifestations. It was confirmed that the sublingual administration of UNAM-HIMFG is a suitable route for immunization through MALT and that, as previously observed [28], it activates the production of TNF-α, which apparently contributes significantly to the resolution of the infectious process. UNAM-HIMFG lysate has a curative effect on CUTI without the need for antimicrobials. This work describes phase two of a pre-clinical study, which gives the guideline to proceed to phase three in individuals with CUTI.

## Figures and Tables

**Figure 1 ijms-25-09876-f001:**
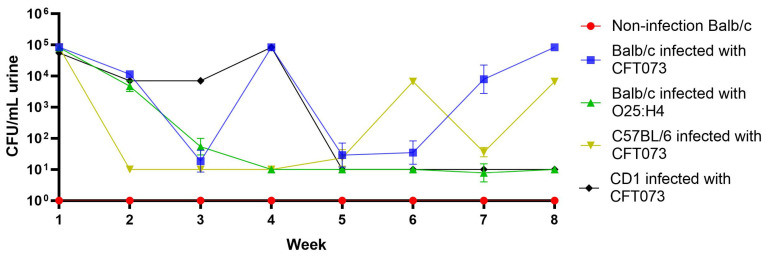
Urinary tract infection induced with UPEC strains in Balb/c, C57BL/6, and CD-1 mice. This study was conducted over eight weeks, analyzing the presence of bacteria in urine.

**Figure 2 ijms-25-09876-f002:**
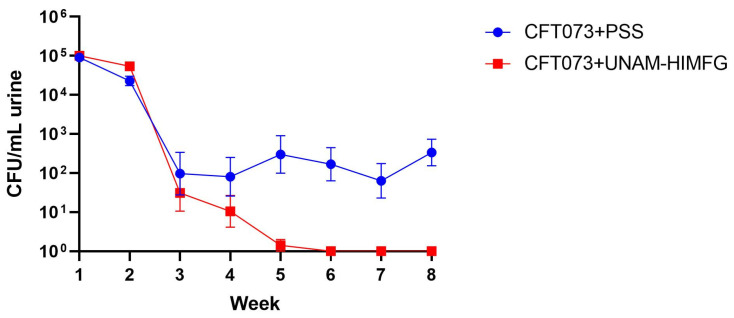
*E. coli* CFT073 recovered from urine samples in Balb/c mice to which it was administered UNAM-HIMFG lysate or PSS. Abbreviations: (CFT073 + PSS) *E. coli* CFT073 infected and PSS administered; (CFT073 + UNAM-HIMFG) *E. coli* CFT073 infected and UNAM-HIMFG lysate treated.

**Figure 3 ijms-25-09876-f003:**
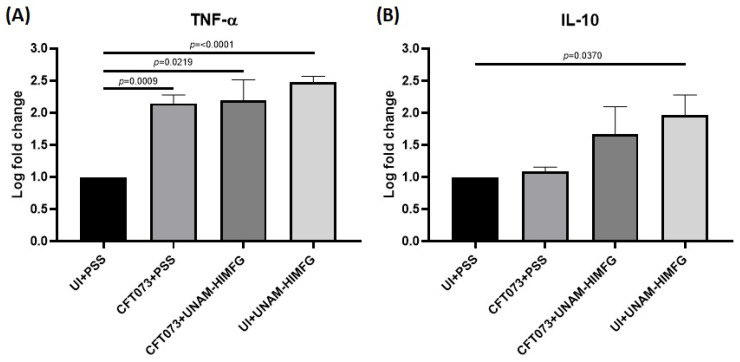
Expression of TNF-α (**A**) and IL-10 (**B**) in mice spleen mRNA of UI + PSS, CFT073 + PSS, CFT073 +UNAM-HIMFG, and UI + UNAM-HIMFG animal groups.

**Figure 4 ijms-25-09876-f004:**
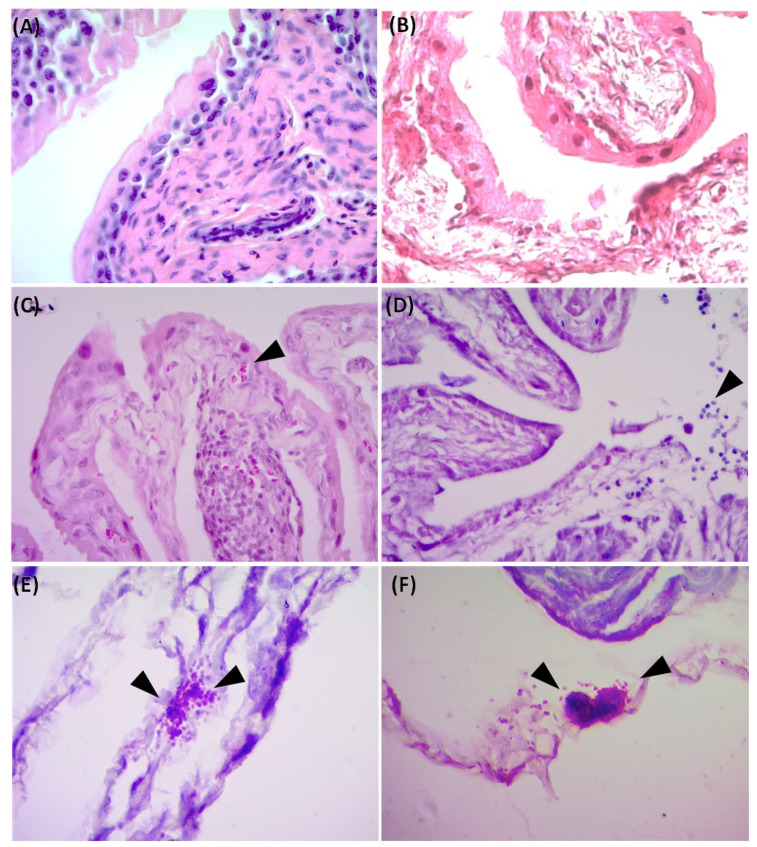
Histological analysis of bladder from Balb/c mice. (**A**) UI + PSS; (**B**) CFT073 + UNAM-HIMFG; (**C**–**F**) CFT073 + PSS; (**C**) The arrow indicates Granulomatous inflammation; (**D**) Arrow shows areas of mild inflammation and leukocytic infiltration; (**E**,**F**) Arrows show Gram-negative bacilli forming colony-like communities. Magnification: 400× (**A**–**D**) and 1000× (**E**,**F**). Hematoxylin/Eosin stain (**A**–**D**); Sandiford stain (**E**,**F**).

**Table 1 ijms-25-09876-t001:** Specific primers to determine somatic and flagellar antigens of O6:K2:H1 and O25:H4 *E. coli* strains.

Targeted Gene	Sequence (5′-3′)	Size (bp)	UPEC Strains	References
*wzy O6* F	GGATGACGATGTGATTTTGGCTAAC	783	CFT073	[63]
*wzy O6* R	TCTGGGTTTGCTGTGTATGAGGC			
*FliC H1* F	ATGCGCTGACTGCATCAAAG	774	CFT073	[62]
*FliC H1* R	CCTTGCCGTTGTTAGCATCG			
*wzy O25* F	GTTCTGGATACCTAACGCAATACCC	229	RMR(U3)02	[63]
*wzy O25* R	AGAGATCCGTCTTTTATTTGTTCGC			
*FliC H4* F	GATTTCAGCGCGGCGAAACT	150	RMR(U3)02	[62]
*FliC H4* R	GGTTGCAGAATCAACGACCG			

**Table 2 ijms-25-09876-t002:** Primers utilized for quantification of TNF-α, IL-10, and GAPDH gene expression.

Gene	Primer	Sequence (5′-3′)	Size (bp)	References
*gpdh*, glyceraldehyde-3-phosphate dehydrogenase [*Mus musculus* (house mouse)]	*gapdh* F	TGGCAAAGTGGAGATTGTTGCC	156	[67]
	*gapdh* R	AAGATGGTGATGGGCTTCCCG		
*Tnf-α*, tumor necrosis factor [*Mus musculus* (house mouse)]	*TNF-a* g	GGTGCCTATGTCTCAGCCTCTT	139	[47]
	*TNF-a* R	GCCATAGAACTGATGAGAGGGAG		
*IL-10* [*Mus musculus* (house mouse)]	*IL-10* F	CGGGAAGACAATAACTGCACCC	130	[67]
	*IL-10* R	CGGTTAGCAGTATGTTGTCCAGC		

## Data Availability

The original contributions presented in the study are included in the article, further inquiries can be directed to the corresponding authors.

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
