# Peer review of "UNAM-HIMFG Bacterial Lysate Activates the Immune Response and Inhibits Colonization of Bladder of Balb/c Mice Infected with the Uropathogenic CFT073 Escherichia coli Strain"

_ijms, 2024, doi:10.3390/ijms25189876_

Round 1

Reviewer 1 Report

Comments and Suggestions for Authors

The authors present a novel experimental study demonstrating the ability of a bacterial lysate to activate the immune response and inhibit the colonization of the bladders of mice infected with uropathogenic E Coli. The study is well designed; the results are impressive; and the manuscript is well-written. This is an important area of investigation, and I commend the authors for their strong work.

In the discussion section, the authors should elaborate on the next research steps and the potential barriers to translating this to the clinical arena. What additional animal experiments are needed (if any) before proceeding with human studies? In practice, would this be a vaccine only for people with recurrent UTIs? Would this potentially have a role in preventing UTIs in patients with chronic indwelling catheters or chronic intermittent catheterization?

Comments on the Quality of English Language

There are several minor grammatical errors. There are also several very long sentences that are difficult to follow. 

Author Response

Dear reviewer, we appreciate the time you took to review the manuscript. Your comments are clear and precise, they were of great help in making improvements to its structure and writing.

What additional animal experiments are needed (if any) before proceeding with human studies?

Although the results presented here are preliminary, these provide elements to consider UNAM-HIMFG lysate as having healing effects in the treatment of UTI without the use of any antimicrobials. However, in this work we were not able to evaluate a group of mice with recurrent infections; therefore, we consider, as a next step, the inclusion of a group of mice treated with antimicrobials and wait for the outcome of the infection. For this, a longer treatment time is required, approximately 60 days is needed to evaluate the recurrent infection events. Afterward, the animals would be administered the UNAM-HIMFG lysate. This treatment scheme would be closer to the observed in human patients.

In practice, would this be a vaccine only for people with recurrent UTIs?

1.- In previous studies, we have been working with monovalent lysates (autolysates, also named autovaccines), these showed outstanding results in cases where chronic infections presented poor response to antimicrobial treatment. (References: (Pathogens 2020, 9, doi:10.3390/pathogens9020102; Microorganisms 2021, 9, 1–13, doi:10.3390/microorganisms9091811 & Front. Public Heal. 2023, 11, doi:10.3389/fpubh.2023.1240392).

2.- With regards to our polyvalent lysate being a treatment for acuate infections, the application of the UNAM-HIMFG lysate is intended for the treatment and control of ongoing UTI and not for the prevention of these infections.  

Would this potentially have a role in preventing UTIs in patients with chronic indwelling catheters or chronic intermittent catheterization?

It is plausible that the UNAM-HIMFG lysate has a role in preventing UTis in patients with chronic indwelling catheters due to the immunostimulant effect and inflammation control herein observed. Although, a study in human patients should be performed in order to fully resolved this matter.

Reviewer 2 Report

Comments and Suggestions for Authors

The article investigates the potential of UNAM-HIMFG lysate as an alternative treatment for chronic urinary tract infections (CUTIs) that can circumvent the issues associated with antibiotic resistance. This study is particularly timely and significant given the increasing global health concern regarding multidrug-resistant bacterial strains. The use of a bacterial lysate to stimulate the immune response and reduce bacterial colonization in a mouse model presents a novel approach to managing UTIs without relying on traditional antibiotics. The concept of using a bacterial lysate as a prophylactic or therapeutic vaccine against UTIs could significantly impact public health.

Areas for Improvement:

  1. Clarity and Structure: The article could benefit from improved structuring and clearer descriptions in some sections to enhance readability and understanding.
  2. Condense the Methodology and Results: Some sections appear overly detailed which might detract from the main findings. Please consider condensing these sections to focus more succinctly on the key methods and results. In addition, please consider shorten the introduction with appears too long and complex. Simplify Technical Language: Where possible, simplify the technical jargon to make the article more accessible to non-specialists. This includes rephrasing complex sentences and reducing the use of acronyms without first defining them.
  3. Statistical Analysis Detail: While the study mentions statistical significance in various comparisons, a more detailed explanation of the statistical methods and clearer presentation of the data could help in validating the results more robustly.
  4. Broader Implications: The discussion could be expanded to better address the broader implications of the findings, including potential scalability, practicality in clinical settings, and any anticipated challenges in transitioning from a mouse model to human trials. In your discussion of alternative treatments for UTIs and their broader implications, it would be beneficial to consider the role of the urinary microbiome as explored PMID: 38298766. Their findings suggest that specific urinary bacteria, identified through comprehensive 16S rRNA sequencing, are not only indicative of bladder cancer but could potentially influence chronic inflammation pathways similar to those you are studying with the UNAM-HIMFG lysate (PMID: 38298766). Such an integration could provide a more holistic view of how microbial interventions might impact bladder health beyond UTIs. Please cite and discuss. 
  5. Discussion: To emphasize the dire need for alternative therapeutic options in the face of rising antibiotic resistance, particularly against carbapenem-resistant Enterobacteriaceae, it would be pertinent to discuss the findings from the review outlined in PMID: 38399502. This review not only details the biological mechanisms behind carbapenem resistance but also highlights the clinical urgency to develop non-antibiotic treatments for complicated urinary tract infections. Incorporating this reference will strengthen the context of your research, illustrating the global challenges of antibiotic resistance and thereby enriching the significance of your findings. While fosfomycin remains an effective first-line treatment for uncomplicated cystitis in women, as evidenced by the study reported in PMID: 37739241, resistance rates, particularly among ESBL-producing bacteria, highlight a critical need for alternative therapies. The findings from this study support the relevance of your research on the UNAM-HIMFG lysate as a potential non-antibiotic treatment that could circumvent these emerging resistance challenges. This will not only enrich the discussion section of your manuscript but also connect your findings to broader issues in antimicrobial resistance, emphasizing the timeliness and potential impact of your work.

Author Response

Dear reviewer, we appreciate the time you took to review the manuscript. Your comments were clear and precise, which was of great help in making improvements to its structure and writing.

Areas for improvement:

Clarity and structure:

1.- The article could benefit from better structuring and clearer descriptions in some sections to improve readability and understanding.

Condense methodology and results: Some sections seem too detailed, which could detract from the main findings. Consider condensing these sections to focus more succinctly on the key methods and results. Also, consider shortening the introduction that seems too long and complex. Simplify technical language: Where possible, simplify technical jargon to make the article more accessible to non-specialists. This includes rephrasing complex sentences and reducing the use of acronyms without first defining them.

The suggestions were considered and the proposed sections were condensed, additionally an attempt was made to shorten the introduction and improve the writing in accordance to the proposed indications.

2.- Statistical analysis details: While the study mentions statistical significance in several comparisons, a more detailed explanation of the statistical methods and a clearer presentation of the data could help to more robustly validate the results.

A thorough revision was made in this regard, and a more detailed description of the statistical analysis was added in the methods section of the manuscript in accordance with the comment.

3.- Broader implications: The discussion could be expanded to better address the broader implications of the findings, including potential scalability, practicality in clinical settings, and any anticipated challenges in transitioning from a mouse model to human trials. In your discussion of alternative treatments for UTIs and their broader implications, it would be beneficial to consider the role of the urinary microbiome as explored in PMID: 38298766. Your findings suggest that specific urinary bacteria, identified through comprehensive 16S rRNA sequencing, are not only indicative of bladder cancer but could potentially influence chronic inflammation pathways similar to those you are studying with the UNAM-HIMFG lysate (PMID: 38298766). Such integration could provide a more holistic view of how microbial interventions might impact bladder health beyond UTIs. Please cite and discuss.

The comment is very valuable, as the discussion included some aspects regarding the potential use of the lysate to improve inflammatory problems associated with changes in the microbiota and in malignant processes such as cancer. It was also mentioned how resistance to several antibiotics promotes changes in sensitive patterns. About scalability, an anticipated challenge in the transition from a mouse model to human trials, we consider that it is still necessary to carry out other trials in the animal model, such as a longer follow-up to find out if, when lysate administration is stopped, recurrent infections occur due to the persistence of the bacteria in the cells. Another trial that we consider important is to administer the lysate before the infection and thus evaluate its availability as a preventive vaccine. These and some other trials on the immune response and signaling pathways will provide more solid elements for the use of the lysate in chronic inflammation problems.

4.- Discussion: To emphasize the need for alternative therapeutic options in the face of increasing antibiotic resistance, particularly against carbapenem-resistant Enterobacteriaceae, it would be pertinent to discuss the findings of the review outlined in PMID: 38399502. This review not only details the biological mechanisms behind carbapenem resistance but also highlights the clinical urgency of developing non-antibiotic treatments for complicated urinary tract infections. Incorporating this reference will strengthen the context of your research, illustrating the global challenges of antibiotic resistance and thus enriching the significance of your findings. While fosfomycin remains an effective first-line treatment for uncomplicated cystitis in women, as demonstrated by the study reported in PMID: 37739241, resistance rates, particularly among ESBL-producing bacteria, highlight a critical need for alternative therapies. The findings of this study support the relevance of your research on UNAM-HIMFG lysate as a potential non-antibiotic treatment that could circumvent these emerging resistance challenges. This will not only enrich the discussion section of your manuscript, but will also connect your findings to broader issues on antimicrobial resistance, emphasizing the timeliness and potential impact of your work.

We tried to identify the intention of the proposal, therefore at the final part of the discussion we included information related to your proposal. We hope we were able to understand the suggestions and that changes in the discussion complies with what was requested.

Round 2

Reviewer 2 Report

Comments and Suggestions for Authors

I concur with the revisions made to the manuscript. The authors have diligently responded to our feedback, resulting in a clearer and more impactful paper. I support its publication as it now stands.